



# Assessing the impact of SSTs on a simulated medicane using ensemble simulations

Robin Noyelle[1], Uwe Ulbrich[2], Nico Becker[2,3], and Edmund P. Meredith[2]

[1]Ecole polytechnique, Route de Saclay, 91128 Palaiseau, France
[2]Institut für Meteorologie, Freie Universität Berlin, Carl-Heinrich-Becker-Weg 6-10
[3]Hans Ertel Centre for Weather Research, Optimal Use of Weather Forecast Branch, Berlin, Germany

**Correspondence:** Robin Noyelle (robin.noyelle@polytechnique.edu)

**Abstract.**

The sensitivity of the October 1996 medicane in the western Mediterranean basin to sea surface temperatures (SSTs) is investigated via 24-member ensembles of regional climate model simulations. Eleven ensembles are created by uniformly changing SSTs in a range of $-4$ K to $+6$ K from the observed field, with a 1 K step. By using a modified phase space diagram and a simple compositing method, it is shown that the SST state has a minor influence on the tracks of the cyclones, but a strong influence on their intensities. Increased SSTs lead to greater probabilities of tropical transitions, to stronger low- and upper-level warm cores, and to lower pressure minima. The tropical transition occurs sooner and lasts longer, which enables a greater number of transitioning cyclones to survive landfall over Sardinia and to re-intensify in the Tyrrhenian Sea. The results demonstrate that SSTs influence the intensity of fluxes from the sea, which leads to greater convective activity before the storms reach their maturity. These results suggest that the processes at steady-state for medicanes are very similar to tropical cyclones.

## 1 Introduction

The Mediterranean basin is one of the most cyclogenetic regions in the world, a majority of its cyclones being baroclinic (Wernli and Schwierz, 2006). However, it has been shown since the 1960s, using remote sensing instruments, that unusual cyclones with a visual similarity to tropical cyclones are also sometimes observed (Ernst and Matson, 1983; Mayengon, 1984; Rasmussen and Zick, 1987). In contrast to asymmetric, synoptic scale extratropical cyclones, these cyclones are of a smaller scale, axisymmetric, frontless and, sometimes, with a well defined eye at their center. These cyclones are now referred to as Tropical-Like Mediterranean Cyclones, or Medicanes. According to the climatological study of Cavicchia et al. (2014a) over the last 60 years, medicanes have a low frequency activity of $1.57 \pm 1.30$ events per year.

Medicanes are intense low pressure systems that present similar characteristics to tropical cyclones: an area with no cloud at the center, spiral bands surrounded by deep convection, intense surface winds, a low- and upper-troposphere warm-core, formation over the sea, and a rapid dissipation after landfall (Pytharoulis, 2018). In recent decades, thanks to high-resolution regional numerical weather prediction models and observational data, detailed examinations of several medicane cases have been carried out (e.g. Reale and Atlas, 2001; Chaboureau et al., 2012; Miglietta et al., 2013; Tous and Romero, 2013; Tous et al., 2013; Picornell et al., 2013; Miglietta et al., 2015; Cioni et al., 2016; Miglietta et al., 2017).



The question of the influence of sea surface temperatures (SSTs) on the development and the characteristics of medicanes has been tackled by several studies. Reed et al. (2001) found a minor influence of small SST anomalies on the medicane of 23 January 1982. Homar et al. (2003) showed that the scenario of development of the medicane of 12 September 1996 had many similarities with the air-sea interaction instability mechanism, especially the latent-heat fluxes acting as a sustainer of convection. According to them, the medicane would have been both weaker and had a delayed development if the SSTs had been colder. Using an axisymmetric cloud-resolving model in which other environmental influences were not included, Fita et al. (2007) showed a strong sensitivity of the same medicane to SSTs. Miglietta et al. (2011) studied the impact of systematic changing of SSTs in a simulation of the 26 September 2006 medicane. According to them, the cyclone was mainly sensitive to uniform SST changes larger than 2 K and increasingly lost its tropical features as the SSTs became colder. Pytharoulis (2018) investigated the influence of SSTs through the imposition of climatological SSTs and uniform warm and cold SST anomalies on the 7 November 2014 medicane. A linear deepening and a longer lifetime of the medicane were observed as the SST anomalies increased from $-3$ K to $+1$ K.

The October 1996 Medicane was previously studied by Mazza et al. (2017), using a 50-member ensemble of regional climate model simulations. It was shown that standard extratropical dynamics were responsible for the cyclogenesis and that the tropical transition of the cyclone resulted from a warm seclusion process. Our present study aims to assess the role of SSTs in the development and the intensity of this cyclone. Ensembles are created for different SST states through dynamical downscaling of reanalysis data and applying a uniform change to the observed SST field in a range of $-4$ K to $+6$ K. To the best of our knowledge, ours is the first study to use ensemble simulations combined with uniform SST changes to evaluate the influence of SSTs on the formation of a particular medicane.

Section 2 describes the model setup and the methodology. The results are presented in section 3, while the discussion and conclusions are included in section 4.

## 2 Data and methodology

### 2.1 Experimental setup

In this paper we study medicane Cornelia, which occurred in the western Mediterranean between 7 and 10 October 1996. The cyclone formed between the Balearic Islands and Sardinia, then moved through Sardinia, the Tyrrhenian Sea and finally dissipated after landfall over Calabria. For a synthesized description of the case see Mazza et al. (2017), and see Reale and Atlas (2001) and Cavicchia and von Storch (2012) for a more extensive report.

The numerical simulations are performed with the full-physics, non-hydrostatic COSMO Climate Limited-Area Model (CCLM; Rockel et al. (2008)) version cosmo4.8-clm19. CCLM is the community model of the German regional climate research community jointly further developed by the CLM-Community. The two-step downscaling configuration consists of a $257 \times 271$ grid point, $0.165°$-resolution parent domain in which a $288 \times 192$ grid point, $0.0625°$-resolution inner domain is nested. For both downscaling steps the model setup includes an extended microphysics scheme accounting for cloud water and cloud ice for grid-scale precipitation based on Kessler (1969), the Ritter and Geleyn (1992) radiation scheme, and the Tiedtke





parametrization scheme for convection (Tiedtke, 1989). Heat fluxes from the ocean to the atmosphere are parameterized using a stability and roughness-length dependent surface flux formulation based on Louis (1979). Both domains feature 40 vertical levels and a 6-hourly update of the boundary conditions. One ensemble is performed, driven by the 0.7° resolution ERA-Interim (Dee et al., 2011).

## 2.2 Ensemble generation

According to previous studies (Davolio et al., 2009; Chaboureau et al., 2012; Cioni et al., 2016; Mazza et al., 2017), numerical forecasts are highly sensitive to initial and boundary conditions. In particular, the location of the lateral boundaries can have a strong impacts on circulation anomalies within the model domain (e.g. Miguez-Macho et al., 2004; Becker et al., under review). In this study, the technique of domain shifting is employed to generate an ensemble (Pardowitz et al., 2016). This technique consists of two downscaling steps with a series of simulations with domains which are slightly shifted in space. We use the same procedure as in Mazza et al. (2017):

- 1st downscaling step:

    - Locate a central domain over Europe

    - Shift the central domain in longitude and latitude in eight directions (north, south, east, west, northeast, northwest, southeast, southwest) in three steps (0.25°, 0.50° and 0.75°)

    - Run the model on each of the 24 domains using ERA-Interim as initial and boundary conditions

- 2nd downscaling step:

    - Use the 24 simulations of the 1st downscaling step as initial and boundary conditions for the smaller, nested domain with a finer resolution over the western and central Mediterranean basin

This results in one set of 24 simulations for the reference SSTs from ERA-Interim. As in Mazza et al. (2017), the initialization time of the first step simulations is 0000 UTC 1 October 1996, while the inner domains are initialized at 0000 UTC 4 October 1996. A 72-h lag was previously found to be a satisfactory compromise between introducing sufficient spread in the ensemble and the ability to capture the event in the simulations (Mazza et al., 2017).

## 2.3 SST changes

To assess the sensitivity of the October 1996 medicane to SSTs, we perturb the SST forcing in the 0.0625° domain by adding SST anomalies in a range of −4 K to +6 K, in 1 K increments, to the observed SST field. In addition to the observed-SST ensemble, this creates 10 extra 24-member ensembles of the case, each with their own unique SST forcing. SSTs were modified uniformly across the entire 0.0625° domain.





## 2.4 Cyclone phase space

We use the three-dimensional diagnostic methodology proposed by Hart (2003), which has already been successfully applied to the study of medicanes (Davolio et al., 2009; Cavicchia and von Storch, 2012; Miglietta et al., 2013; Mazza et al., 2017) to analyze their tropical transition. We use a modified version of the original phase space to take into account the limited vertical
extent and the smaller spatial scale of medicanes compared to tropical cyclones (Picornell et al., 2013; Miglietta et al., 2013; Cioni et al., 2016; Mazza et al., 2017). The three following parameters that define the phase space are computed in a radius of 150 km (as successfully used by Mazza et al. (2017)) :

- The thermal symmetry in the lower troposphere (B): the difference in mean 600-900 hPa thickness between left and right semi-circles with respect to the cyclone's trajectory

- The lower-tropospheric thermal wind $(-\mathrm{V}_T^L)$: the vertical derivative of the cyclone's geopotential height perturbation between 900 and 600 hPa

- The upper-tropospheric thermal wind $(-\mathrm{V}_T^U)$: the vertical derivative of the cyclone's geopotential height perturbation between 600 and 400 hPa

The resulting phase space diagrams can be very erratic at an hourly time scale. Therefore, a 3-hour running mean is applied
to the computed parameters to give the final phase space diagram. Following Hart (2003), in order to classify a cyclone as a medicane the following objective criteria must apply simultaneously: $\mathrm{B} < 10$ m, $-\mathrm{V}_T^L > 0$ and $-\mathrm{V}_T^U > 0$.

## 2.5 Cyclone tracking

The simulated cyclones are tracked based on the mean sea-level pressure (MSLP). As hourly data was used, this proved to be sufficient to correctly follow the cyclone track. The tracking algorithm works as follows:

- Initially, the location of the minimum of pressure is identified within the area of the western Mediterranean basin at 0000 UTC 8 October; at that point in time the cyclone is fully developed with core pressure below 1013 hPa, it is located over Sardinia and is found in every simulation

- The positions of the MSLP minima in the adjacent time steps (backward and forward in time) are determined using a nearest-neighbor algorithm, applied within a circle with a radius of $0.9°$ from the previous MSLP minimum

- The algorithm stops if the cyclone's minimum pressure exceeds 1013 hPa or has made landfall at the French or Spanish coastline (Corsica, Sardinia, Sicilia and continental Italy are not considered as landfall)

- Every track is verified manually with respect to its consistency with the MSLP fields

This algorithm gives coherent trajectories, both for cyclones with and without a tropical transition. The algorithm assures that only one cyclone track is identified in each ensemble member. Continental Italy is not considered as landfall in order



to follow the medicanes even when they make landfall, in particular in order to calculate the evolution of the phase space parameters after landfall. In practice, medicanes dissipate rapidly after landfall.

This method gives reasonable tracks during most of the cyclone's life time, which allows the computation of the phase space parameters. In the early phase of the cyclone the tracks can be slightly erratic. However, this does not affect the results, because

this period is excluded from the analysis. The phase space parameter B could significantly change with a more sophisticated tracking algorithm, because it is sensitive to the exact location of the track position. However, $-\mathrm{V}_T^U$ on its own proved to be sufficient to discriminate between medicanes and non-medicanes.

Track density plots are obtained by counting, for each grid point, the number of tracks that cross a 50 km radius circle around the respective grid point for all members of a given SST ensemble, and dividing by the total number of tracks (see Kruschke

et al. (2016) for a similar method applied to Northern Hemisphere winter storms).

## 2.6 Cyclone compositing

In order to extract the mean signal from fields that show large variability, we use a simple arithmetic averaging technique called cyclone compositing. This technique has been frequently applied to both extratropical and tropical cyclones (e.g. M. Frank, 1977; Bracken and Bosart, 2000; Bengtsson et al., 2007; Catto et al., 2010; Mazza et al., 2017). Following Mazza et al. (2017),

we align the temporal evolution of the simulated cyclones to a common reference time: for each track we identify the time step when $-\mathrm{V}_T^U$ is maximum, $B < 10$ m and $-\mathrm{V}_T^L > 0$. This instant is meant to reflect the stage of maximum warm-core strength, and is hence referred to as the maximum warm-core time (MWCT).

As we focus on transitioning cyclones, we build one composite for each SST state by averaging the 10 transitioning cyclones that have the largest $-\mathrm{V}_T^U$ at MWCT (except when $\Delta\mathrm{SST} = -4$ °C because there only 7 transitioning cyclones occur).

## 3   Results

### 3.1   Probability of transition and track density

With the specified criteria based on the phase space parameters, it is determined if a cyclone encounters a tropical transition, i.e. if it can be regarded as a medicane. This is done for each of the 24 ensemble members of each of the 11 SST settings. It is shown that increasing SSTs leads to an increasing number of medicanes (Fig. 1a). While at $\Delta\mathrm{SST} = -4$ K only 30% of the

ensemble members generate a transitioning cyclone, at +5 K and +6 K all members produce a medicane. The largest increase in medicane development rate is found between -4 K and -3 K (from 7 to 16 medicanes). It could be speculated that this sudden increase is related to a specific SST threshold, similar to the empirical threshold of $26.5$°C, which is found for the development of tropical cyclones (Gray and Brody, 1967). However, here the SSTs are much lower, around $20$ °C.

From each ensemble, those cyclones which are classified as a medicane are selected. For this subset, the mean and standard

deviation of the period of time during which the cyclone was classified as a medicane is calculated. The length of this period increases almost linearly with increasing SSTs (Fig. 1b). For an SST change of -4 K, the period of time is very short with





values of around 5 h. In contrast, the period of time lasts longer at higher SSTs. For example, at $\Delta$SST $= +6$ K the cyclones are classified as medicanes on average for about 104 h. Not only the mean, but also the standard deviation between the time periods in the different ensemble members increases with increasing SSTs.

Figure 2 presents track densities of cyclones for an SST change of -3 K, 0 K, +3 K and +6 K. The figure must be viewed
taking into account that cyclones begin to form between Sardinia and the Balearic Islands, then move through Sardinia and finally go south-east of the Tyrrhenian Sea. All track density plots are very similar when SSTs change and it only seems to be a greater dispersion of tracks when we increased SSTs, especially east of Sardinia and south of continental Italy. Therefore, one can conclude that the SST state has little influence on the track of the medicane, with the large-scale steering flow instead playing the dominant role.

However, figure 3 shows the track densities obtained with the same method, but where tracks are taken into account only when cyclones are classified as medicanes. The two local maxima of density east and west of Sardinia illustrate that the simulations exhibit two distinct classes of medicane formation: those for which MWCT occurs before crossing Sardinia, and which weaken after the crossing, and those for which MWCT occurs after crossing Sardinia, mostly near the Italian peninsula, and continue to deepen after the crossing. It is observed that between 60% to 85% of the medicanes are of the second type,
depending on the SST change. However, there is no systematic dependency between the observed percentage and $\Delta$SST. Fig. 3 also shows that, apart from increasing the period of time when cyclones are transitioning, the track is always similar: first transition before Sardinia, crossing Sardinia then moving south-east to Calabria through the Tyrrhenian Sea. Even if the variability is increasing with higher SSTs, it seems that tracks are not SST-dependent.

## 3.2   Intensity of the cyclones

As a first step, the intensities of the cyclones are analyzed in terms of the minimum core pressure. For this purpose, the composite minimum of mean sea-level pressure (i.e. the mean of minimum pressure along the track of the ten cyclones that have the greatest $-V_T^U$ at MWCT) is calculated for each $\Delta$SST. The composite minimum of MSLP decreases non-linearly from 1004 hPa at $-4$ K to 981 hPa at $+6$ K (Fig. 4). Although the standard deviation, i.e. the variability, becomes very high for an SST change of $+6$ K, it is clear that there is a non-linear relationship between SSTs and the minimum of pressure
of the medicanes. SSTs seem to act as an amplifying factor of an already existing minimum of pressure caused by a higher tropospheric feature.

As the SSTs are not uniformly distributed in the Mediterranean Sea, one may ask whether the differences between the different realizations of the cyclone are caused by differences in the local SSTs along the specific cyclone tracks of the individual ensemble members. To assess the role of local SSTs along the cyclone tracks, we calculate the mean SSTs in a radius of
150 km around the cyclone center and take the average of those values from MWCT$-20$ h until MWCT for each track. These average SST values are presented together with the minimum pressure for all ensemble members of all SST states (including non-transitioning cyclones) in Figure 5. For each SST state, the local SSTs encountered by all cyclones before MWCT vary in a range of 1 K around a mean value. In each ensemble the local SSTs along the cyclone track are thus roughly similar. The minimum of pressure, however, can still vary considerably. For example, at $+6$ K the simulated values of minimum pressure



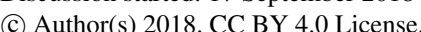


range from 954 hPa to 1002 hPa. This indicates that on the one hand there is a clear tendency to have more intense cyclones at higher SSTs, while on the other hand, the local SSTs are not sufficient to fully account for the variability of the cyclones' minimum pressure.

Figure 6 shows the temporal evolution of the composite minimum of sea level pressure for all $\Delta$SST from $-20$ h before to $+10$ h after MWCT. Every lines are almost equally spaced at $-20$ h but they diverge when MWCT is approaching to reach a minimum at MWCT for almost every SST change. It is evident that at higher SSTs the deepening process of the medicane begins earlier and that the deepening rate is larger than at lower SSTs.

It is observed that the phase space parameters $-V_T^U$ and $-V_T^L$ at MWCT increase linearly with increasing SSTs (Fig. 7). It is obvious that medicanes are getting stronger when SSTs increase, as the thermal winds between 600-900 hPa and 400-600 hPa at MWCT are increasing in almost the same proportion. Fig. 8 presents the temporal evolution of the composite of those two parameters from $-20$ h before to $+10$ h after MWCT for each $\Delta$SST. There is roughly a parallel evolution for each $\Delta$SST for the parameter $-V_T^U$, with similar rates of increase. This contrasts to the evolution of minimum pressure, which showed different rates for different $\Delta$SST (Fig. 6). The evolution for the parameter $-V_T^L$ is more erratic but, at least for positive SST changes, the observation is similar.

## 3.3 The influence of the fluxes from the sea

Similarly to tropical cyclones, the main source of potential energy of medicanes is the thermodynamic disequilibrium between the atmosphere and the underlying sea surface. Therefore, it is generally accepted that there is a direct relationship between SST and cyclone intensity (Miglietta et al., 2011). The prescribed SST changes directly affect the sensible and latent heat fluxes from the ocean into the atmospheric boundary layer. To analyze the heat fluxes under the different SST conditions, the composite structure of the latent heat fluxes from the sea are computed 20 h and 10 h before MWCT, and at MWCT. The horizontal structure of the medicane is remarkably well defined for $\Delta$SST $= +3$ and $+6$ K, with strong gradients of pressure near the center of the cyclone, very low heat fluxes in the eye region and the strongest heat fluxes of more than 400 W/m2 in a radius between 50 and 100 km around the center (Fig. 9). This structure is consistent with the air-sea interaction proposed by Emanuel (1986) for tropical cyclones. It is worth noting that the horizontal structure of the fluxes is not perfectly symmetrical, which is caused by the surrounding islands and continental land area. In particular, there are systematically weaker fluxes north-east of the composite medicane. This effect is the imprint of continental Italy, which is characterized by considerably lower latent heat fluxes than the ocean areas.

For $\Delta$SST $= -3$ K there is no well defined spatial structure in the composite heat fluxes and the heat fluxes are very low with values below 150 W/m2 (Fig. 9, top row). Nevertheless, the cyclones of this composite are classified as a medicane based on the phase space parameters. Therefore, the fluxes from the sea might play a minor role in the formation of those medicanes.

In composites of vertical cross section around the cyclone centers, the vertical wind speeds were analyzed (not shown). It was found that the largest vertical wind speeds occur in the middle troposphere around 500 hPa. Figure 10 shows the 10 h mean of the vertical wind speeds at the 500 hPa level of the composite medicane and its evolution before MWCT. Maximum vertical wind speeds occur around 50 km away from the center of the cyclone, similar to the strongest heat fluxes. This reflects





the development of deep convection in the entire troposphere, i.e. the deepening process which leads to the intensification of the medicane. As in the case of the heat fluxes, the intensity of the vertical wind speeds increases with increasing SSTs.

As one would expect, fluxes from the sea are higher when SSTs increase, leading to greater vertical wind speeds and hence more intense deep convection throughout the troposphere. This in turn leads to a more intense deepening of the medicane and

stronger sea level pressure gradients, thus more intense medicanes. This chain of processes is coherent with the classical model of Emanuel (1986) for the steady-state of tropical cyclones.

## 4   Discussion and conclusions

In this study the sensitivity of a simulated medicane in the western Mediterranean to SSTs was analyzed in an ensemble of full-physics, non-hydrostatic regional model simulations with COSMO-CLM. In a first downscaling step, 24-member ensembles

were created by systematic shifts of the model domain. In a second downscaling step with a horizontal resolution of 0.0625°, SSTs were changed uniformly by adding SST anomalies in a range of -4 K to +6 K, in 1 K increments, to the observed SST field, creating eleven 24-member ensembles.

The cyclones were analyzed and classified according to a modified cyclone phase space following Hart (2003). For each SST change, out of the transitioning cyclones, the 10 cyclones featuring the strongest upper-level warm cores were composited

(except when $\Delta\mathrm{SST} = -4$ K because there are only 7 transitioning cyclones).

It was shown that the number of transitioning cyclones increases with increasing SSTs. A particularly strong increase was found between $\Delta\mathrm{SST} = -4$ K and -3 K, which corresponds to average SST values of around 16 °C within the area of medicane development. These results support the idea that a threshold exists for tropical-like cyclones in the Mediterranean Sea, similar to for tropical cyclones (Gray and Brody, 1967). Miglietta et al. (2011) suggested that, as the presence of medicanes is associated

with cold-air intrusions, they can form even when the SSTs are below the threshold of 26.5 °C for tropical cyclones. Similarly, the duration of the transition increased almost linearly with SSTs. The cyclone tracks showed small random variations in the different ensemble members. However, the tracks of the cyclones, with or without tropical transition, showed no systematic dependency on SSTs compared to the control situation ($\Delta\mathrm{SST} = 0$ K). This suggests that the track characteristics depend rather on the large-scale dynamics of the upper-level low, which determines the environmental steering conditions relevant for

the movement of the medicane.

The intensity of the composite medicane depends strongly on SSTs. Its minimum of pressure decreases when SSTs increase, following a non-linear relationship, in contrast to the intensity of its low- and upper-troposphere warm core, which follow the linear evolution of the duration of the medicane. The process of transitioning is roughly similar when SSTs change, except that the cyclones transition earlier and are less affected by orography when crossing Sardinia. As SSTs increase, the sensible and

latent heat fluxes from the sea into the atmosphere increase. The vertical wind speeds in a region of 50 km around the eye of the medicane also increase, which can be explained by an intensification of deep convection processes caused by the increased heat fluxes. This deep convection leads to deeper sea level pressure minima and stronger pressure gradients and upper-level warm cores, which occur at the same time, just before the medicanes make landfall and begin to dissipate.



Miglietta et al. (2011) found in a similar study of uniform SST changes that the sensible- and latent-heat fluxes from the sea surface during the transit of the cyclone across the Mediterranean Sea have the effect of modifying the boundary layer and are thus efficient mechanisms for convective destabilization. They concluded that warmer (colder) SSTs produce stronger (weaker) sea-surface fluxes, favoring an earlier (delayed) removal of convective inhibition and enhancing (reducing)

the development of convection and the intensification of the cyclone. The results of our study, using ensemble simulations, are in strong agreement with these conclusions. They also concluded that taking $\Delta$SST as a control parameter, the critical value for which the atmospheric circulation displays the appearance of the medicane, having the characteristics of the actually observed phenomenon, is about $\Delta$SST $= -3$ K. Even though we did not compare our results to the observed phenomenon and the situation they studied is located further south than ours, therefore showing higher SSTs of 1 to 2 °C, we also find that a

critical value for the appearance of medicanes is around $\Delta$SST $= -3$ K. Pytharoulis (2018) found an almost linear deepening of the medicane and an increasing life-time as the SST anomalies increased from 3 K to $+1$ K. A non-linearity was, however, found as even warmer SST anomalies were applied: a weaker medicane was simulated and this was mainly attributed to the lack of a well defined upper air warm core. A non-linearity was also found at warmer SSTs in our study, but only for minimum pressure and in the direction of deepening and not weakening.

It is clear that our approach using prescribed SST changes can not fully simulate the impact of SSTs on the development of a medicane, since feedback processes from the medicane to the ocean are not taken into account. In particular, our modeling strategy reflects AMIP-studies, where the complexity of ocean-atmosphere interactions are removed from the modelling process by using prescribed SSTs in global atmospheric general circulation models (Gates, 1992). This allows a clear but rather idealized attribution of the observed effects to the applied SST changes. It should be noted that the inflow into the computational

domain is not changed, thus producing a large differences between SSTs and air temperatures close to the lateral boundaries. Furthermore, strong gradients are also induced between SSTs and land temperatures. However, the temperatures over land, in particular in the coastal areas, proved to adapt rapidly to the SST changes before the formation of the cyclone. Akhtar et al. (2014) studied the robustness of COSMO-CLM coupled with a one-dimensional ocean model (1-D NEMO-MED12). They showed that at high resolution, the coupled model is able to not only simulate most of medicane events but also improves the

track length, core temperature, and wind speed of simulated medicanes compared to the atmosphere-only simulations, suggesting that the coupled model is more proficient for systematic and detailed studies of historical medicane events. It would be valuable to assess the role of SSTs with such a coupled model.

With climate change, in the last decades of the 21st century (2070-2099), the SSTs of the Meditteranean Sea are projected to increase by 1.73 to 2.97 K relative to 1961-1990 (Adloff et al., 2015). Similarly to Pytharoulis (2018), the results of this study

suggest that, if the upper air conditions of the atmosphere remain unchanged, stronger and longer lasting medicanes than today would appear more frequently in the Mediterranean basin. However, the development of medicanes is influenced by additional factors like the presence of baroclinic instability or a cold cut-off low in the upper atmospheric layers. It is likely that climate change will also affect these factors. Romero and Emanuel (2013) and Cavicchia et al. (2014b) both showed that the intensity of medicanes is projected to increase, while their frequency will decrease. Ensemble sensitivity studies, such as that presented

here, are thus a valuable tool for offering insights into how the life-cycles of medicanes may differ in a warmer climate.





*Code and data availability.* Information about the availability of the COSMO-CLM source code can be found on https://www.clm-community.eu. The simulations generated as part of this work are not publicly available. Researchers interested in scientific collaboration and data usage are asked to contact the authors.

*Acknowledgements.* The computational resources were made available by the German Climate Computing Center (DKRZ). The authors
5 would like to acknowledge the European Union for funding this research through an Erasmus+ scholarship. R.N. would like to thank Ingo Kirchner, Stefan Pfahl and Edoardo Mazza for their fruitful comments.



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





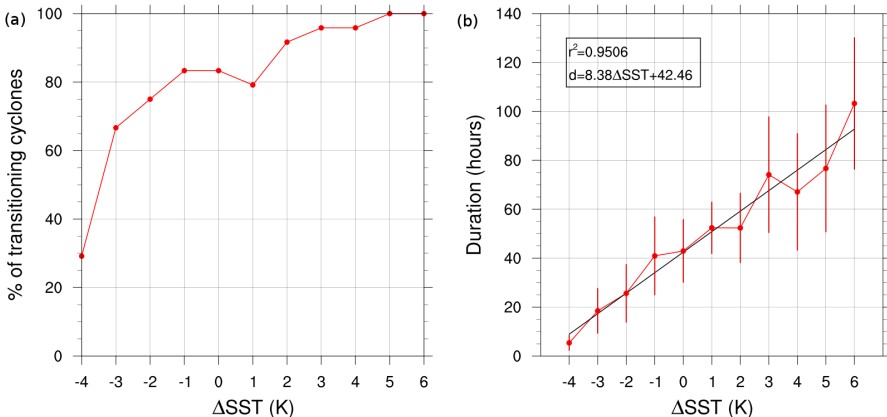

**Figure 1.** (a) Percentage of cyclones encountering a tropical transition over the 24-member ensembles for different SST changes, using classification criteria based on the cyclone phase space. (b) Mean period of time during which transitioning cyclones are classified as medicanes for different SST changes. Error bars represent standard deviation. The black line is based on a linear regression.

**Figures**





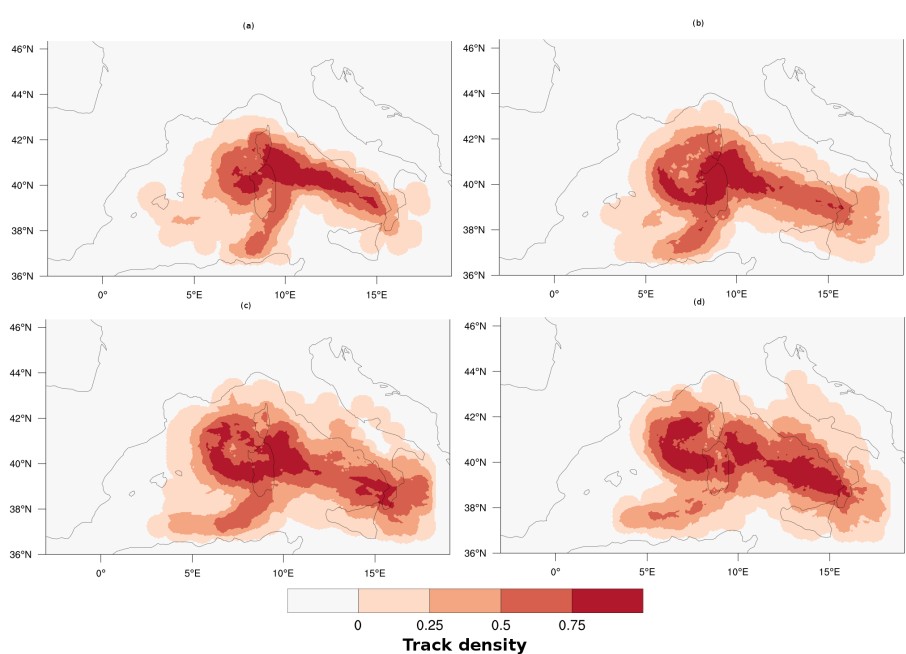

**Figure 2.** Track densities based on all cyclones for an SST change of a) −3 K, b) 0 K (original SSTs), c) +3 K and d) +6 K.

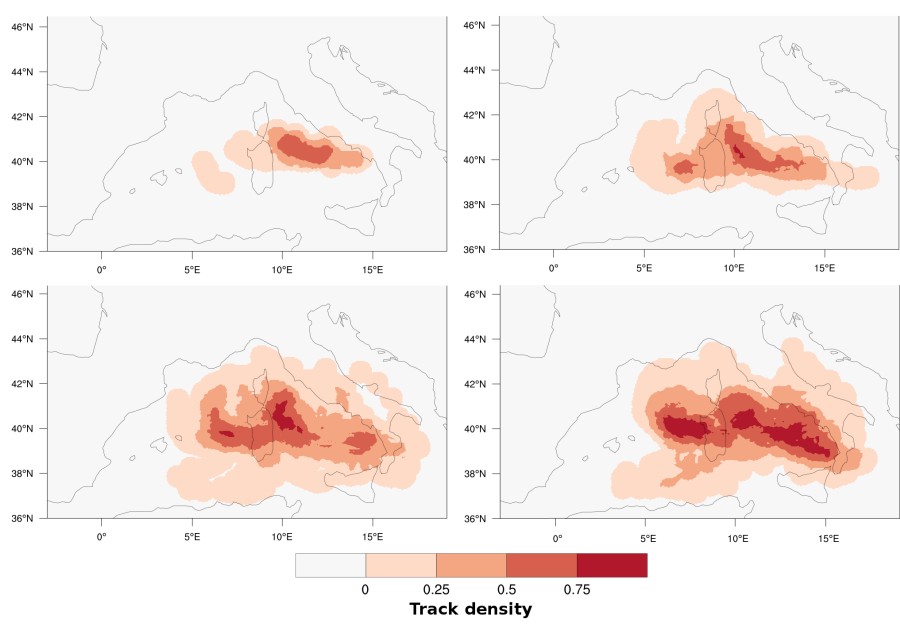

**Figure 3.** Track densities based on all cyclones classified as medicanes for an SST change of a) $-3$ K, b) 0 K (original SSTs), c) $+3$ K and d) $+6$ K.




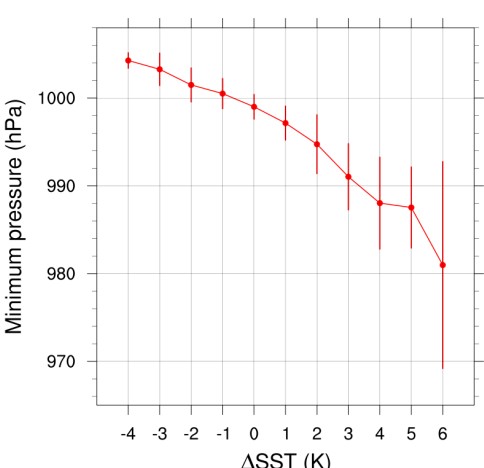

**Figure 4.** Dependence of composite minimum pressure on SST change. The error bars represent standard deviation.





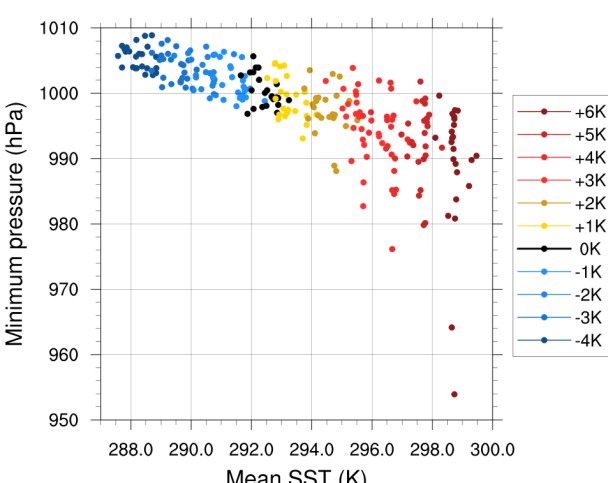

**Figure 5.** Dependence of minimum pressure on mean SST in a radius of 150 km, as encountered by the cyclone from from −20 h before MWCT until MWCT. Colors represent global SST change.





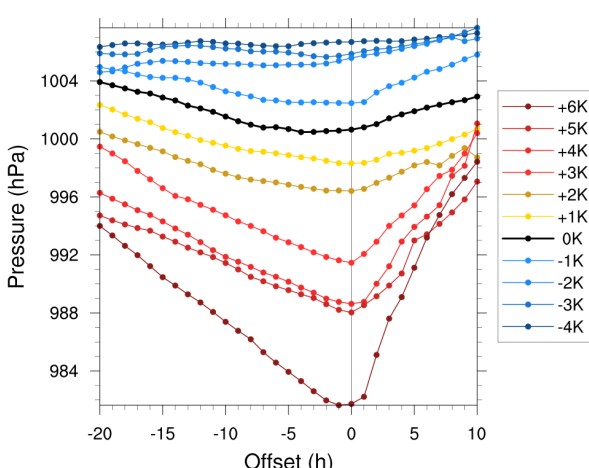

**Figure 6.** Composite time series of minimum pressure from −20 h before until +10 h after MWCT for each SST change.




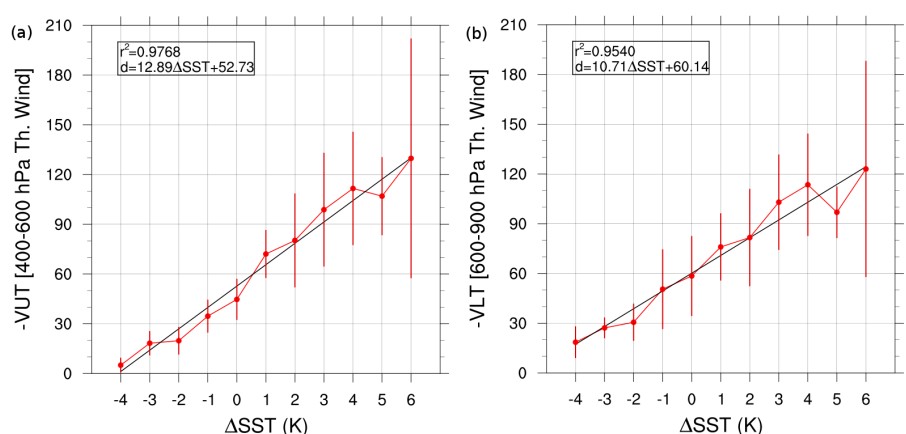

**Figure 7.** Dependence of composite of (a) $-V_T^U$ and (b) $-V_T^L$ at MWCT on SST change. The error bars represent standard deviation.





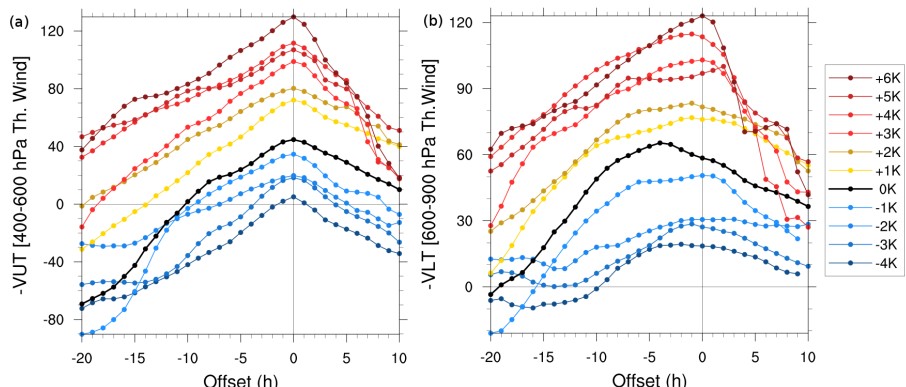

**Figure 8.** Composite time series of (a) $-V_T^U$ and (b) $-V_T^L$ from $-20$ h before to $+10$ h after MWCT for each SST change (for the meaning of each colour see 6).





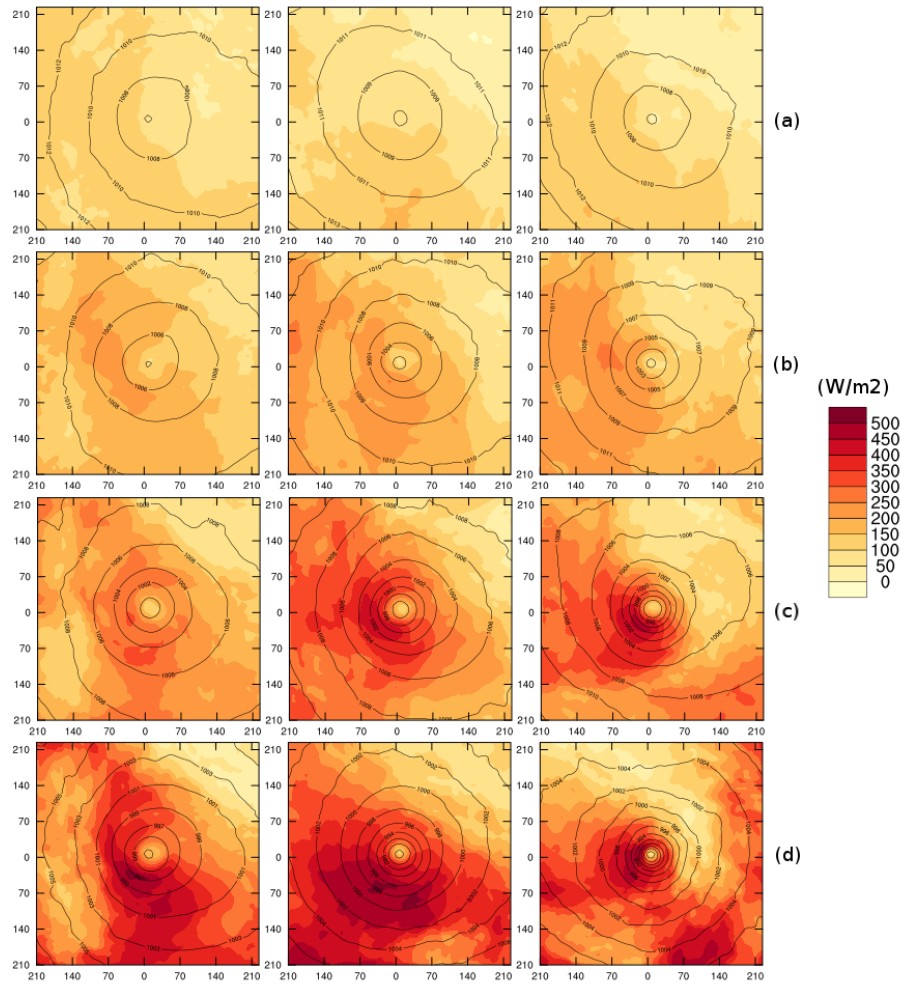

**Figure 9.** Composite latent heat fluxes from the sea for an SST change of a) −3 K, b) 0 K (original SSTs), c) +3 K and d) +6 K (colors) and composite mean sea level pressure (contours) at MWCT−20 h (first column), MWCT−10 h (second column) and MWCT (third column). The values at the x- and y-axis indicate the distance from the center of the medicane in km.





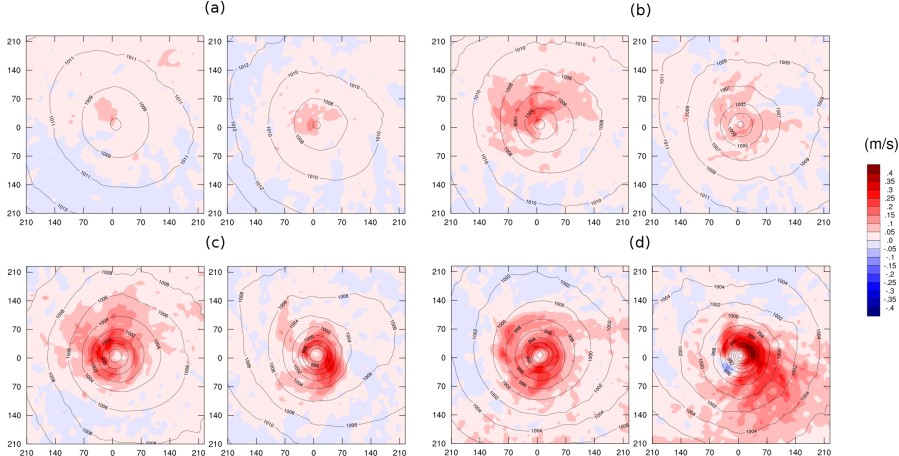

**Figure 10.** 10 h mean of composite vertical wind speed at 500 hPa for an SST change of a) −3 K, b) 0 K (original SSTs), c) +3 K and
d) +6 K (colors) and composite mean sea level pressure (contours) between MWCT−20 h and MWCT−10 h (first column), and between
MWCT−10 h and MWCT (second column). The values at the x- and y-axis indicate the distance from the center of the medicane in km.