# Peer review of "Assessing the impact of SSTs on a simulated medicane using ensemble simulations"

_Natural Hazards and Earth System Sciences, 2018_

## Short Comment (SC1) · 20 Sep 2018

I would like to congratulate with the Authors for the interesting and comprehensive study of the Medicane Cornelia. I am actually performing simulations of the same case (but without changing SST) reaching similar conclusions.

Just a comment: I think that the Authors should make more clear that these simulations have to be interpreted in a statistical sense in the "numerical" world, so may not correspond exactly to the observed cyclone track (which I believe, is a bit southward compared to the track density in Fig. 2b).

Also, numerical simulations with a coupled numerical atmospheric-wave-model of a Medicane have been recently published after Akthar et al. (2014). Ricchi et al. (Sensitivity of a Mediterranean tropical-like cyclone to different model configurations and coupling strategies, Atmosphere, 8, 92, 1-32, 2017) considered the case of November 2011 from a coupling modelling perspective. This paper may be helpful for the conclusive discussion.

best regards

Mario Marcello Miglietta
* * *

---

## Referee Comment (RC1) · Anonymous Referee #1 · 8 Oct 2018

The paper discusses the sensitivity of a simulation of a medicane to uniform changes on sea surface temperature (SST). It clearly shows that the impact of SST is primarily on the intensity of the medicane. The results are well presented and the paper well written. However the paper suffers from a lack of documentation of the medicane and its environment (large-scale flow and SST). The major comments listed below should be considered before publication in NHESS.

**Major comments**

Description of the medicane. This should be done in terms of track and MSLP minimum. A description of the upper-level steering flow and of SST is also needed.

Justification on SST change. The SST field in a range of -4K to +6K. What is the

rationale for changing SST over such a wide range, largely beyond the uncertainty ion the SST measurement?

**Other comments**

Page 4, line 3. The Hart diagram was first applied to medicanes by Chaboureau et al. (2012). It was not used by Davolio et al. (2009)

Page 5, line 24. At ΔSST=0K, only 80% of the ensemble generate a medicane. This suggests that other factors than SST are important for the development of the medicane. It would be worthwhile to comment this result.

Page 6, line 8. As the large-scale steering flow plays a dominant role, it should therefore be documented somehow in a figure.

Page 6, line 21. Please justify why the **composite** minimum of mslp is shown, instead of the median value for example. The same question holds for Figures 6, 7 and 8.

Page 6, line 27. Please plot a map of SST to show its not uniform distribution.

---

## Referee Comment (RC2) · Anonymous Referee #2 · 12 Oct 2018

The paper "Assessing the impact of SSTs on a simulated medicane using ensemble simulations" from R. Noyelle et al. exploits a large ensemble of regional climate model simulations to analyze the impact of varying sea surface temperatures on various properties of a historical medicane case. This topic is interesting an relevant to the ongoing research on medicanes, given the potential implications for both forecasting and better understanding of climate projection for such impactful phenomena.

I find the scientific goals clearly set out, the paper well written, and the provided analysis well supporting the conclusions.

I only have a number of (mostly minor) comments that in my opinion should be addressed prior to publication on NHESS.

GENERAL COMMENTS:

- while reading the paper in several occasions I wondered what the spatial pattern and numerical values of the SST field look like at the time of the medicane occurrence. I think it would be useful if a figure with e.g. the original prescribed SST field could be added.

- one could wonder if the domain shifting technique can potentially introduce some systematic effects (e.g. anomalies in the atmospheric circulation) that could have a comparable role to variations of the SSTs on the evolution of the cyclones. Even if it is not in the scope of this paper, it would be useful if the authors can somewhere in the paper shortly comment on this issue.

SPECIFIC COMMENTS:

PAGE1-LINE2: it is not immediate to understand that the size of the ensemble is 24x11, I suggest to rephrase the sentence to make it more clear that 11 members with perturbed SSTs are produced for each of the 24 domain-shifted ensemble members.

P3-L3: I suggest to move the information on the boundary conditions earlier in the paragraph (e.g. "... resolution parent domain, driven by 0.7 resolution ERA-Interim, in which ..."

P3-L13: in order to provide complete information to the reader, the central domain over Europe should be described or shown (or if it is the same as in Mazza 2017 specify something like "as shown in Fig. xx of Mazza et al. (2017))

P3-L19: same as above for the inner domain

P4-L4: It could be useful for readers not familiar with Hart phase space if it is explicitly mentioned what the modifications with respect to the original phase space are (reduced radius and change of the upper bound for VTU from 300 to 400 hPa)

P5-L5: "this period is excluded from the analysis". Do you mean the "early phase of

the cyclone" is the phase before MWTC-20h? This should be made clearer as at this stage of the paper since you have not specified yet what period of the cyclone lifetime you are going to analyze.

P5-L7: "Vtu on its own proved ...". I find this a bit confusing since there is no universal definition of what a medicane is, and you have stated earlier (Page 4, lines 15,16) that your definition of a medicane involves all the three parameters. Please clarify.

P5-L16: do you find for all the cyclones (also the non transitioning ones) at least one instant where B<10 and -Vtl>0? If not, how do you define MWCT for the cyclones that don't satisfy the condition?

P5-L18: do you find that the the 10 cyclones that enter the composite come more often from some specific 24 domain-shifted ensemble members, or are they uniformly distributed across the ensemble? This is related to general comment 1 above, as it could give some indication on atmospheric patterns that influence the cyclone evolution.

P5-L22: is tropical transition defined as any number of time steps where the cyclone is classified as a medicane or there is some minimum duration of the transition?

P6-L10-19: does the presence of two separate maxima also indicate that the cyclones that have a tropical transition to the west of Sardinia temporarily lose their tropical nature while crossing the island and have a second transition after moving on the sea again?

P6-L24: just looking at the figure, it doesn't look obvious to me - taking into account the error bars - that linearity can be excluded, did you do any statistical test?

P7-L5: "they diverge when MWCT is approaching to reach a minimum at MWCT". The meaning of this sentence is not clear.

Figure 6 and Fig. 8: it seems that the minimum of mslp and the maximum of VTL are reached at MWCT for DeltaSSTs of +1 and larger, but not for negative SST anomalies. Maybe also an indication that the coherent warm core structure is lost when SST is

decreased?

Figure 10: I find it would be clearer if the panels are arranged in two columns, with time increasing from left to right as as in Fig 9

---

## Author Comment (AC1) · 11 Jan 2019

**Response to Reviewer comments on "Assessing the impact of SSTs on a simulated medicane using ensemble simulations".**

R Noyelle, U Ulbrich, N Becker, EP Meredith

January 8, 2019

**1   Preliminaries**

We would like to thank both anonymous reviewers and Mario Miglietta for their comments on our manuscript. We find the comments constructive and appropriate, and think that they will help to improve the manuscript.

In the following pages we set out in detail our responses to the comments and how we plan to act on them.

**2 Response to Anonymous Reviewer #1 (RC1)**

*The paper discusses the sensitivity of a simulation of a medicane to uniform changes on sea surface temperature (SST). It clearly shows that the impact of SST is primarily on the intensity of the medicane. The results are well presented and the paper well written. However the paper suffers from a lack of documentation of the medicane and its environment (large-scale flow and SST). The major comments listed below should be considered before publication in NHESS.*

*Major comments*
*Description of the medicane. This should be done in terms of track and MSLP minimum. A description of the upper-level steering flow and of SST is also needed.*

In a revised version, we will include a (short) new section after the introduction with description and figures of both the observed SST field and the synoptic pattern (including upper-level steering flow) accompanying the medicane; a medicane track will also be included. We shall, however, endeavour to keep this section brief as the medicane in question is already described in detail in Mazza et al. (2017). To avoid unnecessary repetition, we will thus refer the reader to Mazza et al. (2017) where required.

*Justification on SST change. The SST field in a range of -4K to +6K. What is the rationale for changing SST over such a wide range, largely beyond the uncertainty in the SST measurement?*

We must emphasize that it is not our aim to focus on the uncertainty of medicane tracks and intensities under observed conditions, as this is already at least partially been addressed in Mazza et al. (2017). On the contrary, we are more interested in (1) assessing the sensitivity of the medicane to larger SST changes which would be more typically associated with natural variability or anthropogenic climate change, and (2) to try to identify any threshold and/or non-linear behaviour in the system, i.e. how much further must the SSTs be increased (decreased) to guarantee that a (no) medicane will develop. We will state these aims explicitly in a revised version.

*Other comments*
*Page 4, line 3. The Hart diagram was first applied to medicanes by Chaboureau et al. (2012). It was not used by Davolio et al. (2009)*

Thank you for pointing this out. We will correct this statement in the revised version.

*Page 5, line 24. At $\triangle SST = 0K$, only 80% of the ensemble generate a medicane. This suggests that other factors than SST are important for the development of the medicane. It would be worthwhile to comment this result.*

We will comment on this in a revised version. In the submitted version, we already showed (P6, L27-) that the specific local SSTs along the different cyclone tracks are not sufficient to fully account for the variability of the cyclone's minimum pressure; we also attempted to touch on these issues in the Discussion. On the whole, we are of the view that due to the chaotic nature of the weather the formation of medicanes should always be viewed probabilistically rather than deterministically, as even an apparently "ideal" combination of factors cannot guarantee medicane formation. We will endeavour to better communicate the probabilistic nature of medicane formation in a revised version.

*Page 6, line 8. As the large-scale steering flow plays a dominant role, it should therefore be documented somehow in a figure.*

As mentioned above, this will be included in a short new section after the introduction, which will include appropriate new figure(s).

*Page 6, line 21. Please justify why the **composite** minimum of mslp is shown, instead of the median value for example. The same question holds for Figures 6, 7 and 8.*

We thank the Reviewer for this comment, and on re-reading the relevant passage it's apparent that our choice of words could have been clearer. We shall therefore firstly re-formulate the affected passage on page 6 (lines 20-26). In the new passage we will also refer the reader to Section 2.6 of the Methods, where the compositing approach is first explained. The Reviewer rightly points out that the justification of our approach is relevant for a number of figures (e.g. Figs. 4, 6, 7, 8). We will therefore use Section 2.6 as the location for justifying the approach, so that it can be easily referred back to by readers from any point of the manuscript.

As a re-cap: for each SST state, the compositing is performed by first identifying the 10 transitioning cyclones that have the greatest $-V_T^U$ (upper-tropospheric thermal wind) at maximum warm core time (MWCT). These represent the ten strongest medicanes in each 24-member ensemble (recall that there is one 24-member ensemble for each SST state). These ten cyclones are then used as the basis for further analyses. For a given analysis, the same calculation is performed on all ten of the selected cyclones and the result is plotted as the mean of all ten calculations. Figure 4 therefore shows the mean of the 10 pressure minima from each of the 10 identified cyclones (not the minimum of the 10 pressure minima), for each SST state.

The reason for adopting this particular compositing approach is that the definition of when a cyclone is a medicane is imperfect and not universally the same, and can also vary depending on how the Hart diagramme is calculated. It is therefore more instructive to focus on the most powerful medicanes to learn about the true behaviour of transitioning cyclones. This methodology can also be applied identically irrespective of the SST state. Studying the mean instead of the median of the ten selected cyclones is simply one choice among a number of choices which one must make. Nevertheless, we will also repeat the analysis using the median of the ten instead of the mean. If we detect a notable difference we will assess and comment on the role of outliers in the final composite mean values.

*Page 6, line 27. Please plot a map of SST to show its not uniform distribution.*

As mentioned above, this will be included in a short new section after the introduction, with appropriate new figure(s).

**3 Response to Anonymous Reviewer #2 (RC2)**

*The paper "Assessing the impact of SSTs on a simulated medicane using ensemble simulations" from R. Noyelle et al. exploits a large ensemble of regional climate model simulations to analyze the impact of varying sea surface temperatures on various properties of a historical medicane case. This topic is interesting an relevant to the ongoing research on medicanes, given the potential implications for both forecasting and better understanding of climate projection for such impactful phenomena. I find the scientific goals clearly set out, the paper well written, and the provided analysis well supporting the conclusions. I only have a number of (mostly minor) comments that in my opinion should be addressed prior to publication on NHESS.*

**GENERAL COMMENTS:**
*- while reading the paper in several occasions I wondered what the spatial pattern and numerical values of the SST field look like at the time of the medicane occurrence. I think it would be useful if a figure with e.g. the original prescribed SST field could be added.*

This comment echoes sentiments expressed by the other reviewer. As mentioned above, we will include a short new section after the introduction including description and figures of the observed SST field and synoptic situation (including steering flow) which accompanied the medicane. We shall, however, aim to keep this brief as the medicane is already discussed in detail in Mazza et al. (2017). Where necessary, we will refer the reader to Mazza et al. (2017).

*- one could wonder if the domain shifting technique can potentially introduce some systematic effects (e.g. anomalies in the atmospheric circulation) that could have a comparable role to variations of the SSTs on the evolution of the cyclones. Even if it is not in the scope of this paper, it would be useful if the authors can somewhere in the paper shortly comment on this issue.*

It is important to emphasize that all domain-shifts, including the central (unshifted) domain, are equally valid and that there is no "correct" domain position. Any "anomalous" circulation thus introduced to the system is within the range of inherent uncertainty due to imperfect observations (i.e. reanalysis) and the chaotic nature of the system. We will better emphasize this point in the revised methods section, also taking into account what results from the comparison in response to the comment about P5-L18.

**SPECIFIC COMMENTS:**
*PAGE1-LINE2: it is not immediate to understand that the size of the ensemble is 24x11, I suggest to rephrase the sentence to make it more clear that 11 members with perturbed SSTs are produced for each of the 24 domain-shifted ensemble members.*

Thanks for pointing this out. We will rephrase it as suggested in a revised version.

*P3-L3: I suggest to move the information on the boundary conditions earlier in the paragraph (e.g. "... resolution parent domain, driven by 0.7 resolution ERA-Interim, in which ..."*

We agree that this will improve the text and will adopt the suggested rearrangement in a revised version. Thanks!

*P3-L13: in order to provide complete information to the reader, the central domain over Europe should be described or shown (or if it is the same as in Mazza 2017 specify something like "as shown in Fig. xx of Mazza et al. (2017)")*
*P3-L19: same as above for the inner domain.*

Thanks for the suggestion. The domains are indeed the same as in Mazza et al. (2017) and we will thus refer the reader to this paper.

*P4-L4: It could be useful for readers not familiar with Hart phase space if it is explicitly mentioned what the modifications with respect to the original phase space are (reduced radius and change of the upper bound for VTU from 300 to 400 hPa)*

Thank you. We will mention this in a revised version.

*P5-L5: "this period is excluded from the analysis". Do you mean the "early phase of the cyclone" is the phase before MWTC-20h? This should be made clearer as at this stage of the paper since you have not specified yet what period of the cyclone lifetime you are going to analyze.*

Thank you. The Reviewer is correct that here we are referring to the early stages of the cyclone development, which is a long time before MWCT-20h. We will reformulate this in a clearer manner in a revised version.

*P5-L7: "Vtu on its own proved ...". I find this a bit confusing since there is no universal definition of what a medicane is, and you have stated earlier (Page 4, lines 15,16) that your definition of a medicane involves all the three parameters. Please clarify.*

Here we were just trying to explain that at any time when $-V_T^U > 0$ for a given cyclone, that we also always had $|B| < 10m$ and $-V_T^L > 0$ at the same time. Therefore, using $-V_T^U > 0$ as the sole discriminating parameter would have led to the same cyclones being identified. We will more clearly re-word this in a revised version, along the following lines: *While our criteria to distinguish a medicane from a non-medicane system requires that all criteria mentioned in Section 2.4 are fulfilled simultaneously, it turns out that for the systems considered evaluation of $-V_T^U$ alone would have led to the same systems being identified.*

*P5-L16: do you find for all the cyclones (also the non transitioning ones) at least one instant where $B < 10$ and $-Vtl > 0$? If not, how do you define MWCT for the cyclones that don't satisfy the condition?*

We do find for all cyclones (also the non-transitioning ones) at least one instant where $|B| < 10m$ and $-V_T^L > 0$. The MWCT is thus defined for all cyclones as the instant where $|B| < 10m$, $-V_T^L > 0$ and $-V_T^U$ is maximum, as in Mazza et al. (2017). We will mention this fact in the revised version. For the composite plots (Figs. 4, 6-10), we anyway only include cyclones which transition to medicanes, so these figures wouldn't be affected were the above not the case. For Figure 5, however, we include all cyclones (also non-transitioning ones), so for Figure 5 the above is certainly relevant. We will thus also add further clarification to the caption of Figure 5.

*P5-L18: do you find that the the 10 cyclones that enter the composite come more often from some specific 24 domain-shifted ensemble members, or are they uniformly distributed across the ensemble? This is related to general comment 1 above, as it could give some indication on atmospheric patterns that influence the cyclone evolution.*

We will perform this analysis and include the results in the Methods and/or Discussion section, as appropriate based on the outcome.

*P5-L22: is tropical transition defined as any number of time steps where the cyclone is classified as a medicane or there is some minimum duration of the transition?*

We analyze the medicane criteria at hourly intervals. If the criteria are met at one of these intervals then tropical transition is considered to have occured, so there is no minimum number of time-steps. We will mention this in the revised Methods section.

*P6-L10-19: does the presence of two separate maxima also indicate that the cyclones that have a tropical transition to the west of Sardinia temporarily lose their tropical nature while crossing the island and have a*

*second transition after moving on the sea again?*

In our simulations we have not found any cases in which cyclones re-transition east of Sardinia, having lost medicane status over Sardinia. Cyclones which lost their medicane status did not re-gain it. We shall however investigate this in more detail and comment on it further in the Results section.

*P6-L24: just looking at the figure, it doesn't look obvious to me - taking into account the error bars - that linearity can be excluded, did you do any statistical test?*

For the revised version, we will perform a statistical test and, if necessary, modify the language in the manuscript based on the outcome of the test.

*P7-L5: "they diverge when MWCT is approaching to reach a minimum at MWCT". The meaning of this sentence is not clear.*

Thank you for pointing this out. We will reword along the following lines: "The curves are almost all equally spaced at -20 h, but they diverge as MWCT is approaching, and reach a minimum at MWCT for almost all SST states".

*Figure 6 and Fig. 8: it seems that the minimum of mslp and the maximum of VTL are reached at MWCT for DeltaSSTs of +1 and larger, but not for negative SST anomalies. Maybe also an indication that the coherent warm core structure is lost when SST is decreased?*

Figures 6 and 8 are produced based on composites of the strongest ten cyclones (from each SST state) which were classified as medicanes based on the objective medicane classification criteria. In particular, to be classified as medicanes all cyclones need to have a thermal symmetry (**B**) within an acceptable range indicative of tropical transition and this thermal symmetry criterion must also be met for MWCT to be defined. Having said that, being classified as a medicane at some point along its track ($-V_T^U > 0$, $|B| < 10m$, $-V_T^L > 0$) does not guarantee that the cyclone is still a medicane at MWCT ($-V_T^U$ maximum, $|B| < 10m$, $-V_T^L > 0$). Indeed, as stated in our response to the comment about P5-L16, we do find for all cyclones (even non-transitioning cyclones) at least one instant where MWCT can be defined. As suggested by the reviewer, it may be the case that some of the low-SST cyclones which met the medicane criteria at some point along their tracks were no longer medicanes at MWCT. We will further investigate this possibility by (i) considering the fraction of the cyclones in each ensemble which retain medicane status at MWCT and (ii) the coincident temporal evolutions of $-V_T^U$ and $-V_T^L$ in the medicanes from the warmer and colder SST ensembles. Guided by our findings, we will then comment on the point raised by the reviewer in the revised manuscript, as appropriate based on the outcome of our analysis.

*Figure 10: I find it would be clearer if the panels are arranged in two columns, with time increasing from left to right as as in Fig 9*

We will rearrange the panels as suggested in a revised version.

**4   Response to Short Comment #1 (Mario Marcello Miglietta)**

*I would like to congratulate with the Authors for the interesting and comprehensive study of the Medicane Cornelia. I am actually performing simulations of the same case (but without changing SST) reaching similar conclusions.*

*Just a comment: I think that the Authors should make more clear that these simulations have to be interpreted in a statistical sense in the "numerical" world, so may not correspond exactly to the observed cyclone track (which I believe, is a bit southward compared to the track density in Fig. 2b).*

*Also, numerical simulations with a coupled numerical atmospheric-wave-model of a Medicane have been recently published after Akthar et al. (2014). Ricchi et al. (Sesitivity of a Mediterranean tropical-like cyclone to different model configurations and coupling strategies, Atmosphere, 8, 92, 1-32, 2017) considered the case of November 2011 from a coupling modelling perspective. This paper may be helpful for the conclusive discussion.*

*best regards*

*Mario Marcello Miglietta*

We thank the contributor for his comments. We will attempt to make this clearer when introducing the ensemble technique in the methods section, which also relates to comments made by the second reviewer about the domain-shift technique potentially introducing systematic effects. We also thank the contributor for the references, which we will include in the text as appropriate.

**References**

E. Mazza, U. Ulbrich, and R. Klein. The tropical transition of the october 1996 medicane in the western mediterranean sea: A warm seclusion event. *Monthly Weather Review*, 145(7):2575–2595, 2017.

---

## Author Response (AR1)

**Response to Reviewer comments on "Assessing the impact of SSTs on a simulated medicane using ensemble simulations".**

R Noyelle, U Ulbrich, N Becker, EP Meredith

March 11, 2019

**1 Preliminaries**

We would like to thank both anonymous reviewers and Mario Miglietta for their comments on our manuscript. We find the comments constructive and appropriate, and think that they have improved the manuscript.

In the following pages we set out in detail our responses to the comments and how we have acted on them.

**2 Response to Anonymous Reviewer #1 (RC1)**

The paper discusses the sensitivity of a simulation of a medicane to uniform changes on sea surface temperature (SST). It clearly shows that the impact of SST is primarily on the intensity of the medicane. The results are well presented and the paper well written. However the paper suffers from a lack of documentation of the medicane and its environment (large-scale flow and SST). The major comments listed below should be considered before publication in NHESS.

**Major comments**

Description of the medicane. This should be done in terms of track and MSLP minimum. A description of the upper-level steering flow and of SST is also needed.

We thank the reviewer for this suggestion. As mentioned in our initial response (AC1), a synthesized description of the medicane case is already provided in Mazza et al. (2017). There are additionally detailed descriptions of the medicane in Reale and Atlas (2001) and Cavicchia and von Storch (2012). To avoid repitition of the aforementioned works while also recognising the valid comments of the reviewer, we have therefore added a brief overview of medicane Cornelia in a new Section 2, which includes an accompanying new figure. Our new Section 2 contains a decription of Cornelia in terms of its sea-level pressure, track, steering flow and accompanying SSTs. These parameters are all also represented in the new Figure 1. Within the new Section 2 readers are referred to the aforementioned studies for a more detailed synoptic discussion. Figure 1 covers the spatial extent of the inner simulation domain and thus also serves to better illustrate the experimental design.

Justification on SST change. The SST field in a range of -4K to +6K. What is the rationale for changing SST over such a wide range, largely beyond the uncertainty in the SST measurement?

We have added an explaination of our rationale to the penultimate paragraph of the Introduction (P2 L19-23), making clear that the SST range is chosen based on our interest in exploring (1) the sensitivity of the medicane to larger SST changes which would be more typically associated with natural variability or anthropogenic climate change, and (2) to try to identify any threshold and/or non-linear behaviour in the system.

**Other comments**

Page 4, line 3. The Hart diagram was first applied to medicanes by Chaboureau et al. (2012). It was not used by Davolio et al. (2009)

Thank you for pointing this out. We have corrected this statement in the revised manuscript and also removed the reference to Cavicchia and von Storch (2012) (P4 L22-24).

Page 5, line 24. At  $\triangle SST = 0K$ , only 80% of the ensemble generate a medicane. This suggests that other factors than SST are important for the development of the medicane. It would be worthwhile to comment this result.

We have commented on this result in Section 4.1 of the results (P6 L20-24). We now highlight that observed SSTs do not guarantee medicane formation and emphasize that (i) medicane formation should be considered probabilistically, (ii) the observed medicane was just one of many potential realizations for observed SST, and (iii) that small changes in initial conditions can propogate to inhibit medicane formation in the real and model worlds. In addition to this, we also touch on this theme in Section 4.2 (P8 L5-9).

Figure 1: As in Figure 5 of the manuscript ("Dependence of composite minimum pressure on SST change."), except also showing the composite median (black) alongside the composite mean (red).

Page 6, line 8. As the large-scale steering flow plays a dominant role, it should therefore be documented somehow in a figure.

As mentioned above, this has been included in a figure which accompanies the new Section 2, after the introduction.

Page 6, line 21. Please justify why the **composite** minimum of mslp is shown, instead of the median value for example. The same question holds for Figures 6, 7 and 8.

We have firstly re-worded the relevant sentences on (P7 L22-24) to try and better explain what we are doing. As the reviewer comment is relevant to a number of figures, i.e. all those based on compositing, we have secondly used Section 3.6 to explain the rationale behind our approach (P6 L11-14). We have thirdly confirmed that the results are not sensitive to whether we chose the composite mean or median, and duly mention this in Section 3.6 (P6 L13-14); see Figure 1.

Page 6, line 27. Please plot a map of SST to show its not uniform distribution.

As mentioned above, this has been included in a figure which accompanies the new Section 2, after the introduction.

**3 Response to Anonymous Reviewer #2 (RC2)**

The paper "Assessing the impact of SSTs on a simulated medicane using ensemble simulations" from R. Noyelle et al. exploits a large ensemble of regional climate model simulations to analyze the impact of varying sea surface temperatures on various properties of a historical medicane case. This topic is interesting an relevant to the ongoing research on medicanes, given the potential implications for both forecasting and better understanding of climate projection for such impactful phenomena. I find the scientific goals clearly set out, the paper well written, and the provided analysis well supporting the conclusions. I only have a number of (mostly minor) comments that in my opinion should be addressed prior to publication on NHESS.

**GENERAL COMMENTS:**

- while reading the paper in several occasions I wondered what the spatial pattern and numerical values of the SST field look like at the time of the medicane occurrence. I think it would be useful if a figure with e.g. the original prescribed SST field could be added.

We thank the reviewer for this suggestion. As promised in our initial response (AC1), we have included a short new section (Section 2) after the introduction which provides a brief decription of the medicane, while referring the reader to previous studies for more detail. A new Figure 1 accompanies this section, in which the observed SSTs at the time of the medicane are plotted, alongside the sea-level pressure field, cyclone track, and steering flow.

- one could wonder if the domain shifting technique can potentially introduce some systematic effects (e.g. anomalies in the atmospheric circulation) that could have a comparable role to variations of the SSTs on the evolution of the cyclones. Even if it is not in the scope of this paper, it would be useful if the authors can somewhere in the paper shortly comment on this issue.

We have addressed this point by adding clarifying text to Section 3.2 of the Methods ("Ensemble generation", P4 L5-12), which now empasizes that all domain-shifts, including the central (unshifted) domain, are equally valid and that there is no "correct" domain position. Any "anomalous" circulation thus introduced to the system is within the range of inherent uncertainty due to imperfect observations (i.e. reanalysis) and the chaotic nature of the system. In addition to this change, please also see the actions taken (below) in response to the comment about P5-L18 (below), which are relevant for addressing the above comment.

**SPECIFIC COMMENTS:**

PAGE1-LINE2: it is not immediate to understand that the size of the ensemble is 24x11, I suggest to rephrase the sentence to make it more clear that 11 members with perturbed SSTs are produced for each of the 24 domain-shifted ensemble members.

Thanks for pointing this out. We have added an extra sentence at the beginning of the abstract to make clear that there are 11 SST states and 24 members per SST state (P1 L2-3).

P3-L3: I suggest to move the information on the boundary conditions earlier in the paragraph (e.g. "... resolution parent domain, driven by 0.7 resolution ERA-Interim, in which ..."

We have moved the mention of ERA-Interim towards the beginning of Section 3.1 (P3 L11-13). The mention of ERA-Interim from the end of this paragraph has thus been removed and the downscaling procedure is now described over two sentences as follows: "The two-step downscaling configuration begins with a  $257 \times 271$  grid point,  $0.165^{\circ}$ -resolution parent domain which is forced at the lateral boundaries by ERA-Interim reanalyis (Dee et al., 2011). A  $288 \times 192$  grid point,  $0.0625^{\circ}$ -resolution domain is then nested within the  $0.165^{\circ}$  parent domain."

P3-L13: in order to provide complete information to the reader, the central domain over Europe should be described or shown (or if it is the same as in Mazza 2017 specify something like "as shown in Figure xx of

Mazza et al. (2017)") P3-L19: same as above for the inner domain.

Thanks for the suggestion. The domains are indeed the same as in Mazza et al. (2017). We have used the new Figure 1 (SSTs, SLP, etc.) to also show the reader the full spatial extent of the inner domain. In the revised version we thus refer the reader to this figure as appropriate in Sections 3.1 and 3.2. For the larger domain over Europe, we now state in Section 3.1 that this (i) covers an area slightly smaller than the EURO-CORDEX domain, and (ii) is shown in Mazza et al. (2017) (P3 L13-15). In Section 3.2 we now also state that the larger domain over Europe is the same as that in Mazza et al. (2017) (P3 L29).

P4-L4: It could be useful for readers not familiar with Hart phase space if it is explicitly mentioned what the modifications with respect to the original phase space are (reduced radius and change of the upper bound for VTU from 300 to 400 hPa)

We have added this information to the second paragraph of Section 3.4 ("Cyclone phase space", P4 L3-6).

P5-L5: "this period is excluded from the analysis". Do you mean the "early phase of the cyclone" is the phase before MWTC-20h? This should be made clearer as at this stage of the paper since you have not specified yet what period of the cyclone lifetime you are going to analyze.

We have addressed this issue by stating that the earliest period of analysis is 20 hours before the time of maximum warm-core strength and further adding a reference to the following section on cyclone compositing, where MWCT is first defined (P5 L25-27).

P5-L7: "Vtu on its own proved ...". I find this a bit confusing since there is no universal definition of what a medicane is, and you have stated earlier (Page 4, lines 15,16) that your definition of a medicane involves all the three parameters. Please clarify.

We have added the following sentence to the Section 3.4 (formerly Section 2.4): While our method to distinguish a medicane from a non-medicane requires that all criteria mentioned in Section 3.4 are fulfilled simultaneously, for the systems considered in this study the evaluation of  $-V_T^U$  alone would have led to the same systems being identified. With this new sentence (P5 L29-31), we hope to more clearly explain that at any time when  $-V_T^U > 0$ for a given cyclone, that we also always had |B| < 10m and  $-V_T^L > 0$  at the same time. Therefore, using  $-V_T^U > 0$ as the sole discriminating parameter would have led to the same cyclones being identified.

P5-L16: do you find for all the cyclones (also the non transitioning ones) at least one instant where B < 10and -Vtl > 0? If not, how do you define MWCT for the cyclones that don't satisfy the condition?

We do find for all cyclones (also the non-transitioning ones) at least one instant where |B| < 10m and  $-V_T^L > 0$ . We have mentioned this fact in the revised sub-section on cyclone compositing (P6 L10-11). As mentioned in the initial author comments (AC1), for the composite plots (Figs. 5, 7-11) we anyway only include cyclones which transition to medicanes, so these figures are unaffected. For Figure 6, however, we include all cyclones (also non-transitioning ones), so for Figure 6 we have added further text to the caption to state that the analysis is based on all cyclones.